# Absence of bulk charge density wave order in the normal state of UTe$_2$

C. S. Kengle[1,2] ✉, J. Vonka [3], S. Francoual [4], J. Chang [5], P. Abbamonte[2], M. Janoschek [5,6], P. F. S. Rosa [1] & W. Simeth [1,5,6] ✉

A spatially modulated superconducting state, known as pair density wave (PDW), is a tantalizing state of matter with unique properties. Recent scanning tunneling microscopy (STM) studies revealed that spin-triplet superconductor UTe$_2$ hosts an unprecedented spin-triplet, multi-component PDW whose three wavevectors are indistinguishable from a preceding charge-density wave (CDW) order that survives to temperatures well above the superconducting critical temperature, $T_c$. Whether the PDW is the mother or a subordinate order remains unsettled. Here, based on a systematic search for bulk charge order above $T_c$ using resonant elastic X-ray scattering (REXS), we show that the structure factor of charge order previously identified by STM is absent in the bulk within the sensitivity of REXS. Our results invite two scenarios: either the density-wave orders condense simultaneously at $T_c$ in the bulk, in which case PDW order is likely the mother phase, or the charge modulations are restricted to the surface.

Unconventional superconductors not only provide a platform for the investigation of exotic pairing mechanisms beyond electron-phonon coupling[1–4], but also underpin existing technology such as powerful superconducting (SC) magnets or have potential for future applications such as topological quantum computing with fractionalized excitations[5]. In particular, most conventional $s$-wave superconductors are topologically trivial, whereas unconventional superconductors may be topological depending on the underlying SC order parameter. For any newly-discovered unconventional superconductor, the determination of its SC order parameter therefore becomes a central question. However, this task is rendered difficult due to experimental discrepancies, disorder, and intertwined orders, such as charge-density waves (CDWs), magnetism, pair-density waves (PDWs), and nematicity[6,7].

UTe$_2$ is a recent addition to the family of unconventional superconductors[8,9], and its SC order parameter continues to defy consensus. Though there is strong evidence that UTe$_2$ is a spin-triplet superconductor[10–13], reports of chiral, multicomponent, and topological superconductivity[14–17] have been challenged[18–22]. Recently, scanning tunneling microscopy (STM) identified SC order parameter components, $\Delta_{\mathbf{q}}(\mathbf{R})$, that are spatially modulated at three wave-vector components, $\mathbf{q}_i$ ($i = 1$, 2, 3), which are around the Brillouin zone boundary[23] and which display different field and temperature variations[23]. This pair-density wave (PDW) is inherently linked to CDW order, $\rho_{\mathbf{q}}(\mathbf{R})$, which displays the same modulation components as $\Delta_{\mathbf{q}}(\mathbf{R})$ but shifted by a phase difference $\pi$[23–25].

Notably, CDW order survives to temperatures well above the SC critical temperature, $T_c \approx 1.6$ K. This intriguing result may point to two possible scenarios of intertwined order (cf. ref. 24): either uniform SC and PDW orders are dominant and generate subordinate charge modulations or uniform SC and CDW orders dominate and give rise to pair density modulations. In the first case, a superconductor of gap maximum around 250 μeV coupled to a 10 μeV PDW state must induce a CDW with a much larger energy scale of order 25 meV. In the second case, the normal-state CDW (25 meV) coexists with a uniform SC component (250 μeV) below $T_c$ to generate a 10 μeV PDW state at the same wavevectors. While Gu et al. [24] argue that the second scenario (of subordinate PDW order) is more plausible, Aishwarya et al. [23] reason

[1]Los Alamos National Laboratory, Los Alamos, NM, USA. [2]Department of Physics and Materials Research Laboratory, University of Illinois Urbana-Champaign, Urbana, IL, USA. [3]Laboratory for X-ray Nanoscience and Technologies, Paul Scherrer Institute, Villigen PSI, Switzerland. [4]Deutsches Elektronen-Synchrotron (DESY), Hamburg, Germany. [5]Physik-Institut, Universität Zürich, Zürich, Switzerland. [6]Laboratory for Neutron and Muon Instrumentation, Paul Scherrer Institute, Villigen PSI, Switzerland. ✉e-mail: ckengle@lanl.gov; wsimeth@lanl.gov

that a subordinate CDW order is the only plausible scenario and not inconsistent with a normal state CDW. Whether the PDW is the mother order or a subordinate order therefore remains unsettled.

Though STM is a powerful surface probe that provides both phase and domain sensitive information with high accuracy[26,27], it is not inherently sensitive to ordered states in the bulk and therefore unable to make conclusive statements on the penetration depth of a state. In fact, signatures of CDWs observed in STM are equally likely to arise from bulk or surface states, and the two forms of order may coexist in a material[28]. Therefore, with all experimental evidence for either CDW or PDW modulations in UTe₂ being restricted to surface sensitive measurements, it remains unclear whether these modulations in the SC state extend into the bulk and, if so, whether the broadening of the momentum space CDW peaks above $T_c$ is also a bulk phenomenon. Signatures of a phase transition when CDW order vanishes are absent in transport or thermodynamic measurements[29-31], therefore emphasizing the necessity to perform a systematic search for charge-density wave signatures using microscopic bulk-sensitive methods.

Here we provide the desired bulk measurements of the charge order in UTe₂ via resonant elastic X-ray scattering (REXS) performed just above the SC transition ($T = 2.2$ K), which is about four times lower than the highest temperature the CDW in UTe₂ is reported to exist[25]. Resonant diffraction can increase intensity of charge order by more than three orders of magnitude compared to non-resonant scattering[32,33], where intensities are frequently prohibitively weak and which is largely insensitive to weak forms of charge-order not involving lattice degrees of freedom. Therefore, having used incident X-ray energies of 4.95 keV (Te $L_1$ edge) and 3.73 keV (U $M_4$ edge) is central for the conclusion of our study on the absence of bulk charge order in the normal state of UTe₂.

Because STM above the SC transition identified nanometer-sized patches of charge order, we investigated a region in reciprocal space near two of the reported wave vectors with respect to broad signals of correlation lengths smaller than typical patch sizes. Additionally, to consider a state that has a different correlation length in the bulk, we investigated the vicinity of one of the ordering vectors on a tight grid sufficiently dense and large enough to identify resolution limited CDW-signals using incident X-ray energy of 3.73 keV. Our main result is that in both cases the structure factor of the putative CDW is absent from the bulk within our detection limits posed by resonant X-ray diffraction.

## Results

### Surface charge order observed in STM

We first revisit the nature of charge order identified previously in STM studies[23-25]. Data were taken on (011) planes of UTe₂. The top layer after cleaving consists of Te-atoms that form a two-dimensional orthorhombic Bravais lattice with a rectangular centered unit cell (cf. Supplementary Materials Text 1 and Fig. S1). Figure 1a illustrates schematically characteristic signatures in two-dimensional reciprocal space, as seen in Fourier transformed STM images and discussed in refs. 23-25. The periodic arrangement of Te-atoms results in relatively strong structural Fourier-components (black circles). Below $T_c$ long-range charge-order is observed with components $q_1 = (0, 0.57)$, $q_2 = (1, 0.43)$, and $q_3 = (-1, 0.43)$, shown in Fig. 1a in terms of orange circles. With STM being restricted to atomic length scales and extending, at best, over two atomic layers, these studies were, however, unable to directly identify modulation components along (011).

We will now consider putative bulk charge order with wave-vectors that result in these three surface projections. Although a single wave-vector (single-**q** with **q** = (0.57, 0, 0)) lacking a modulation along (011) could on its own lead to this pattern of projections below $T_c$, it is unable to account for the multi-component behavior observed through field and thermal variations (cf. ref. 23). We will therefore consider three independent components **q**₁, **q**₂, and **q**₃ in the three-dimensional Brillouin zone of UTe₂ that possess the projections q₁, q₂, and q₃.

As noted in ref. 23 and illustrated in Fig. 1a, in two-dimensional reciprocal space the projected CDW positions are near the projected high symmetry points W, L₁, and L₂, shown in terms of purple circles and labeled with the same lowercase letters. In three dimensional momentum space, these three high-symmetry points are at the Brillouin-zone boundary and feature a modulation period of two crystallographic units along a ⟨011⟩ axis. Due to the similarity of the geometric shapes that the triangles △q₁q₂q₃ and △wl₂l₁ enclose with the x-axis (orange and purple pentagons) as well as their coinciding location in two-dimensional reciprocal space, we calculated **q**₁, **q**₂, and **q**₃ assuming that they are indeed located close to W, L₁, and L₂ in momentum space with a modulation of two crystallographic units along ⟨011⟩ (see Suplementary Information).

Figure 1 (b) illustrates a section of the 3D Brillouin zone for UTe₂. The relevant characteristic high-symmetry points and calculated CDW wave vectors are show in purple and orange, respectively. Other high-

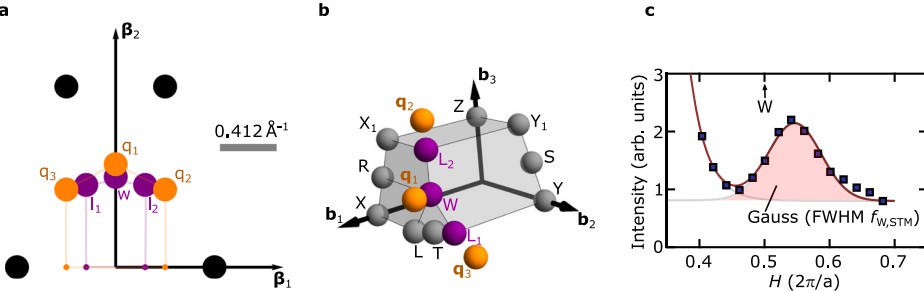

**Fig. 1 | Multi-component surface-CDW identified by means of scanning tunneling microscopy (STM). a** Schematic view explaining the signatures in fast Fourier transformed STM images, as taken on (011) surfaces of UTe₂ below the superconducting transition. The **β**₂-axis is parallel to the crystallographic (100) axis of UTe₂ and **β**₁ is perpendicular to (100) and (011). Black circles indicate the Fourier components due to the rectangular lattice of Te-atoms. Large orange circles correspond to charge-ordered peaks, q₁, q₂, and q₃. The purple circles l₁, w, and l₂ correspond to projections of high-symmetry points (denoted by the respective capital letters) of the three-dimensional Brillouin zone onto the two-dimensional reciprocal lattice. The orange and purple lines show the pentagon shapes that Δq₁q₂q₃ and Δwl₂l₁ enclose with the x-axis. The tiny orange and purple circles correspond to the projections of the large circles onto the **β**₁-axis. The gray bar indicates the size of one r.l.u. along the horizontal axis. **b** High-symmetry points in

the first Brillouin zone of UTe₂, as represented by spheres. Ordering vectors were reported in the vicinity of W, L₁, and L₂ (purple spheres). The orange points correspond to the location of charge-ordered peaks within the (011) plane at the Brillouin zone boundary that have the projections that are shown in (**a**) and that feature a modulation of two crystallographic units along (011). **c** Exemplary cut along the line WS above the superconducting transition. Shown in terms of square symbols is intensity inferred from a fast Fourier transformation of STM picture taken at 4.9 K (cf. Fig. 2b of ref. 25). For the purposes in this study, we fitted the data by a superposition of exponentially decaying background and a Gaussian peak with FWHM $f_{STM}$ (brown lines in lower opacity). The superposition of these two curves is shown in full opacity. The Gaussian profile is shown in orange shading. The location of the W-point is indicated by the black arrow.

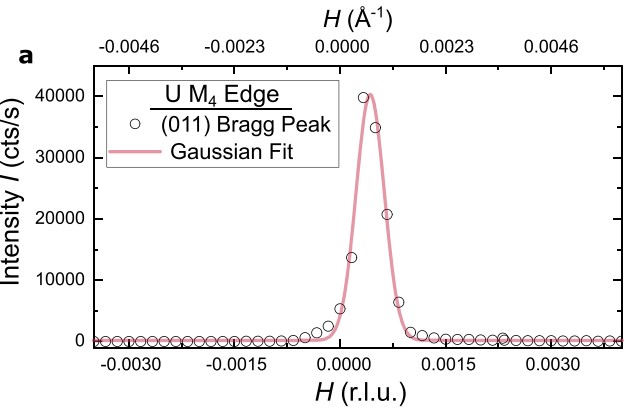

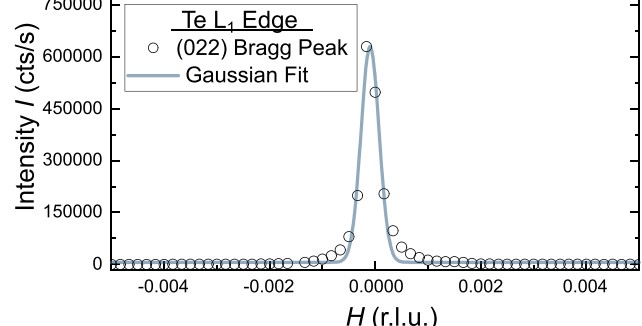

**Fig. 2 | Resolution-limited structural Bragg peaks. a** Structural Bragg peak (011) as recorded with 3.73 keV incident energy and fitted with a Gaussian profile (red solid line) of Full Width at Half Maximum $4.6985 \times 10^{-4}$ r.l.u. $= 6.85 \times 10^{-4}$ Å$^{-1}$.

**b** Structural Bragg peak (022) as recorded with 4.95 keV incident energy and fitted with a Gaussian profile (blue solid line) of Full Width at Half Maximum $3.07 \times 10^{-4}$ r.l.u. $= 4.68 \times 10^{-4}$ Å$^{-1}$. Diffraction data points are represented by circles.

symmetry points are indicated with gray spheres. The CDW wave-vector positions do not match any obvious commensurate value, but are in the plane perpendicular to $\Gamma$S ($\Gamma$ is the origin). The first wave-vector, $\mathbf{q}_1 = (0.57, 0.5, 0.5)$, is slightly off the W point, $\mathbf{q}_2 = (0.430, 0.338, 1.339)$ and $\mathbf{q}_3 = (0.430, 0.662, -0.339)$ are close to the Brillouin zone corners $L_2$ and $L_1$.

Above the SC transition temperature the charge order begins to melt, but persists in the form of patches which shrink upon increasing temperature[34]. For example, at 4.8 K the patches are of the order 5 nm in size. Finally, we use exemplary STM data above $T_c$ to infer a conservative estimation of the expected full width at half maximum (FWHM) of bulk CDW order. Figure 1c shows a typical intensity *vs.* momentum curve obtained from a Fast Fourier Transformation of STM images taken at 4.9 K along the line connecting the points W and S, i.e., between $(\frac{1}{2}, \frac{1}{2}, \frac{1}{2})$ and $(\frac{1}{2}, \frac{1}{2}, 0)$ as specified in conventional orthorhombic coordinates. A pronounced peak near the W point arises from charge-density wave component $\tilde{\mathbf{q}}_1 \approx (0.545, 0.5, 0.5)$ and is characterized by a broad Gaussian profile centered at $H = 0.545$ r.l.u. The tilde denotes momentum transfers and reciprocal space positions that were derived from STM data above the superconducting transition, where charge-density wave order forms nanometer-sized patches (Fig. 1c).

Consider now that the peak FWHM seen in STM represents a convolution of intrinsic peak shape ($f_0 = 2/\kappa_0$), associated with the correlation length $\kappa_0$ of CDW order, and the experimental resolution of the STM instrument, ($f_{STM}$). Assuming that both $f_0$ and $f_{STM}$ describe the widths of Gaussian profiles, we find that the profile in Fig. 1c has a FWHM $f_{W,STM} = \sqrt{f_0^2 + f_{STM}^2}$. The experimental profile therefore serves as an upper bound for the internal width of charge-order, yielding $f_{max} := f_{W,STM} = 0.1439$ Å$^{-1} > f_0$. In turn, a lower bound may be obtained considering that coherent CDW correlations were observed in terms of patches of less than $\kappa_1 = 5$ nm diameter, implying the bound for the intrinsic peak width of $f_{min} := 2/\kappa_1 < f_0$, cf. ref. 35. Combined, these limits on the FWHM can be written $f_{min} = 0.04$ Å$^{-1} < f_0 < 0.1439$ Å$^{-1} = f_{max}$.

## Resonant X-ray scattering
To study bulk microscopic properties of charge modulations in UTe$_2$, we utilized resonant elastic scattering of hard, linearly polarized X-rays and searched for signatures of a multi-component CDW around the Brillouin zone boundary. With penetration depths in UTe$_2$ exceeding 200 nm, these experiments are sensitive to the structure factor of long-range correlations across several hundred crystallographic unit cells.

For the putative resonant enhancement of charge-density wave order, the precise structure factor is unknown. But it is well established, that in the non-resonant case when using linearly polarized X-rays diffraction intensities from charge-order may have maximum intensity in the polarization channel $\sigma\sigma'$, i.e., with both incident and scattered polarization parallel to the scattering plane,[36]. Therefore, in order to maximize the chance to observe charged-ordered Bragg peaks we maintain incident polarization parallel to the scattering plane[36].

In our experiments, structural Bragg peaks measured on UTe$_2$ were limited by experimental resolution represented by Gaussian profiles, indicating ideal crystalline quality of our sample. Figure 2 shows representative scans through (a) the (011) Bragg peak using an incident energy of 3.73 keV and (b) the (022) Bragg peak using an incident energy of 4.95 keV. See Supplementary Information, Text S3, for more details on the setup of our diffraction experiments.

## Survey for putative CDW order in bulk
To search for the CDW in the bulk, we will consider two limits. First, that the CDW peaks are resolution limited (having the same width as the structural Bragg peaks). Next, that the CDW has correlation lengths of order 5 nm, inspired by previous STM measurements[25].

We start with the search for resolution-limited charge-order peaks. Such long-range order has correlation lengths much larger than $2/f_{res}$, where $f_{res}$ denotes the width in momentum space of resolution-limited structural Bragg peaks. To increase our chance to discover charge-order and to avoid missing its signatures due to the high resolution of REXS, we chose an incident energy of 3.37 keV, where the resolution volume is larger compared to 4.95 keV.

We surveyed reciprocal space in the search of signal at $\mathbf{Q}_{1,a}^{CDW} = (-0.57, 2.5, 2.5)$, corresponding to CDW domain of wavevector type $\mathbf{q}_1$, shown by the gray sphere in the center of Fig. 3a. The letter $a$ denotes the specific momentum transfer where the CDW peak was studied in our experiments. To account for possible uncertainties in momentum transfers due to the accuracy of the diffractometer (see Supplementary Information Text S4), we performed a 3-dimensional grid search at positions near $\mathbf{Q}_{1,a}^{CDW}$. The mesh consists of $H$-scans at given $K_0$, $L_0$-coordinates indicated in Fig. 3 in terms of spheres. The spacing along $K$ and $L$ was chosen as 0.0018 r.l.u apart, which is smaller than the instrument resolution (green ellipsoid in Fig. 3a). The step width in each linescan along $H$, $\Delta H = 3.6 \cdot 10^{-4}$ Å$^{-1}$, is smaller by a factor of two than the experimental resolution determined for a nearby specular Bragg peak (c.f. Supplementary Material Text 4), and therefore small enough to account for uncertainties in the incommensurate component of $\mathbf{q}_1$ along (100). The scan range along $H$ was chosen large

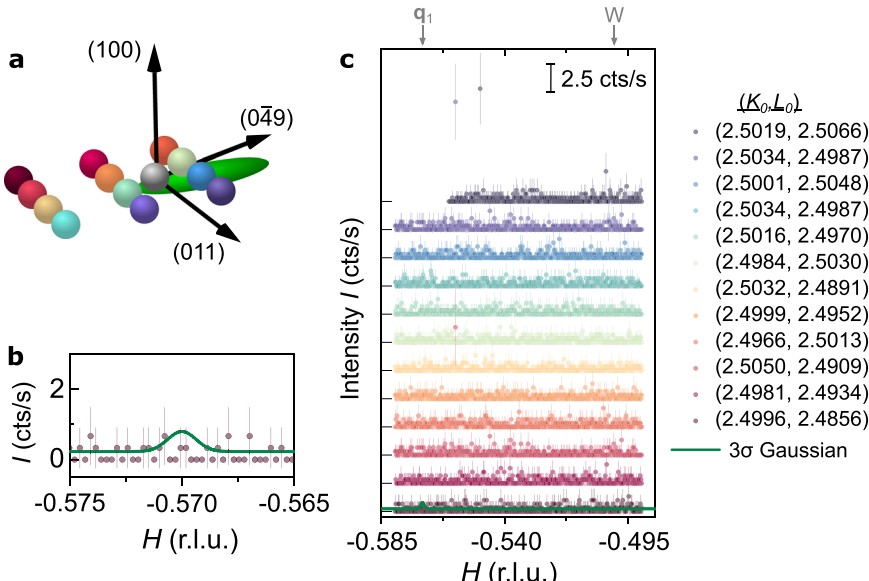

**Fig. 3 | Absence of resolution-limited CDW Bragg peaks near $q_1$. a** Reciprocal space region that was investigated by means of $H$-scans. $K_0$ and $L_0$ coordinates were held constant during each scan and are illustrated in terms of colored spheres. The gray sphere denotes the W-point, close to which the CDW-peak is expected according to STM. The green ellipsoid illustrates the resolution of the diffractometer. **b** $H$ scan at $(K, L) = (2.4996, 2.4856)$, zoomed in to emphasize the shape of a resolution-limited Gaussian of height $3\sigma$ i.e., where $\sigma$ denotes the uncertainty of the background obtained from counting statistics. Error bars denote the uncertainty from X-ray counting statistics. **c** REXS intensity of all $H$-scans performed. Colors denote the $K_0, L_0$-coordinates in (**a**). Plots are offset for clarity, and each horizontal line indicates zero for the corresponding plot. The intensity scale of 2.5 cts/s is indicated by the black scale bar and defines the y-axis. Error bars denote the uncertainty from counting statistics. The location of the points W and $q_1$ is indicated by gray arrows.

enough to account for variations in the modulation component along (100) of the order -0.05 r.l.u. Such variations are observed when moving from the surface into the bulk and may be sample-dependent[32,37].

Figure 3c shows the resulting scans where the color of the plot corresponds to the $K_0, L_0$-coordinates in (a). The scans can be modeled by only a constant background of 0.23(1) cts/s and with averaged standard error from Poisson counting statistics of $\sigma = 0.19$ cts/s. As a conservative upper limit of the maximum intensity due to charge modulations, we assume that one would be able to detect signatures that are three error bars above the background (in the following denoted "$3\sigma$"). As such a signal was not observed (green curve), we conclude that the maximum CDW amplitude at 3.73 keV incident energy is below our $3\sigma$ limit: $A(\mathbf{Q}_{1,a}^{\text{CDW}}) \leq 0.61$ cts/s. Comparing to the peak height of a nearby Bragg peak, (0,1,1), we find that the CDW peak amplitude on the Uranium M-edge is at least five orders of magnitude lower $A_{\text{CDW}} \leq 1.5 \cdot 10^{-5} \cdot A_{(0, 1, 1)}$.

Turning now to the second scenario, where CDW peaks have ~nm-sized short correlation lengths, we expand our search to look for signal at the corresponding bulk components $\tilde{\mathbf{q}}_1$ and $\mathbf{q}_3$. For wavevector type $\mathbf{q}_1$, data were collected in two Brillouin zones using two incident energies: $\tilde{\mathbf{Q}}_{1,a}^{\text{CDW}} = (-0.545, 2.5, 2.5)$ at 3.73 keV and $\tilde{\mathbf{Q}}_{1,b}^{\text{CDW}} = (-0.545, 3.5, 3.5)$ at 4.95 keV. Here $b$ labels another momentum-transfer, where we studied the respective wave-vector. For wavevector of type $\mathbf{q}_3$, scans were taken in one Brillouin zone at $\mathbf{Q}_{3,b}^{\text{CDW}} = (-0.430, 2.338, 2.661)$ with incident energy of 4.95 keV.

The results of these scans for both incident energies and wave-vectors are presented in Fig. 4. Panels (a) and (b) show scans for $\tilde{\mathbf{q}}_1$ at 3.73 keV and 4.95 keV, respectively, and panel (c) shows scans for $\tilde{\mathbf{q}}_3$ at 4.95 keV. The background intensities were each fitted using a line with a constant slope and offset, shown in blue. The fit results are displayed above each panel. Mean error bars, $\sigma$, as obtained from Poisson counting statistics are given by (a) 0.17 cts/s, (b) 8.9 cts/s, and (c) 8.38 cts/s.

A charge ordered peak having a FWHM of $(f_{\min} + f_{\max})/2$ and amplitude $\geq 3\sigma$ would be detectable above the background, as indicated

by the green Gaussian curves in Fig. 4. We conclude that such Gaussian profiles are absent within our detection limit in all three cases.

Comparison of intensities lets us conclude that the maximum intensities of CDW peaks having short correlation lengths in bulk on the Te L-edge are at least eight orders of magnitude smaller than for the structural Bragg peaks. Compared to that, in the cuprate YBa$_2$Cu$_3$O$_{6.67}$[38] even for non-resonant x-rays charge order signals are only seven orders of magnitude smaller relative to strong nearby structural Bragg peaks.

## Upper boundary on atomic displacements

Charge modulations probed by Thomson scattering of X-rays can arise either from spatial modulation of valence electrons or from periodic atomic displacements[39]. These two effects typically appear together and cause each other. With periodic lattice displacements leading to a much stronger response in X-ray scattering, we search for them directly and are further interested in them as indirect evidence of valence modulations[38]. Here, we will calculate upper bounds of atomic displacements in UTe$_2$ based on the scattering intensities reported above. Neglecting any putative resonant enhancement of charge diffraction, the actual displacements of atoms due to charge order are much lower than the values identified here.

Using standard expressions, harmonic displacements of atoms along a direction $\mathbf{d}$ away from the equilibrium position $\mathbf{R}_i$ may be written as $\mathbf{R}_i' = \mathbf{R}_i + u\hat{\mathbf{d}} \cdot \sin(\mathbf{q}_0 \cdot \mathbf{R}_i)$. In diffraction, this leads to charge-order Bragg peaks around structural Bragg peaks $\mathbf{G}$, appearing in leading order at $\mathbf{G} \pm \mathbf{q}_0$.

Neglecting higher order terms, the structure factor of the modulated crystal around $\mathbf{Q} = \mathbf{G}$ can be written as

$$S_{\text{m}}(\mathbf{Q}) = s(\mathbf{G})\delta(\mathbf{Q} - \mathbf{G}) + s(\mathbf{G})\left(\frac{\mathbf{Q} \cdot \mathbf{u}}{2}\right)^2 \delta(\mathbf{Q} - \mathbf{G} \pm \mathbf{q}_0). \quad (1)$$

Here, $S(\mathbf{G}) = s(\mathbf{G}) \, \delta(\mathbf{Q}-\mathbf{G})$ denotes the structure factor around $\mathbf{G}$ of the undistorted crystal. As the structure factor is proportional to the

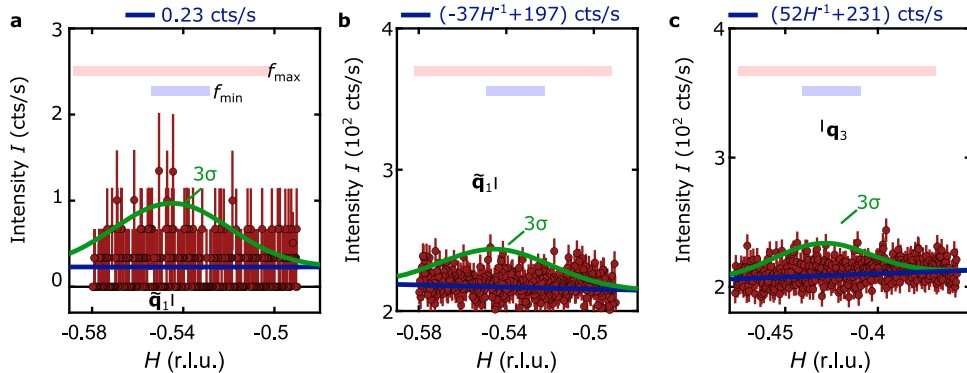

**Fig. 4 | Absence of diffuse charge-scattering peaks in resonant X-ray diffraction data.** The three panels show momentum-space cuts through positions, where according to STM reports diffuse charge scattering peaks of a multi-component CDW above the superconducting transition were reported. Shown in (**a**) is an $H$ scan through $\tilde{\mathbf{q}}_1$ at $(H, 2.5, 2.5)$ using 3.73 keV incident energy, in (**b**) through $\tilde{\mathbf{q}}_1$ at $(H, 3.5, 3.5)$ using 4.95 keV incident energy, and in (**c**) through $\mathbf{q}_3$ at $(H, 2.338, 2.661)$ using incident energy 4.95 keV. As explained in the text and derived from STM, a diffuse CDW signal may be expected to have full width have maximums in the interval $f_{min} = 0.04\,Å^{-1} < f_0 < 0.1439\,Å^{-1} = f_{max}$, as indicated by the blue and red horizontal bars. The red circles present X-ray counts, where the error bars correspond to uncertainties arising from Poisson counting statistics. Points at zero intensity are here shown with vanishing error bars. The background was fitted by linear functions given at the top of each panel (blue thick). The green line corresponds to a Gaussian peak with an amplitude of $3\sigma$ above the background, where $\sigma$ denotes the mean error bar of the background amplitude. The location of the calculated bulk wave-vectors is indicated by black bars.

integrated intensities of Bragg peaks, i.e., $I(\mathbf{Q}) \propto S_m(\mathbf{Q})$, the ratio of intensities of a structural Bragg peak to a charge order Bragg peak therefore yields for the amplitude $\mathbf{u} = u\mathbf{d}$ of the modulation around $\mathbf{G}$

$$\frac{I(\mathbf{q}_0 + \mathbf{G})}{I(\mathbf{G})} = \left(\frac{\mathbf{u} \cdot (\mathbf{q}_0 + \mathbf{G})}{2}\right)^2 \qquad (2)$$

The intensity at a given position in momentum space may arise from a combination of different charge-order satellites associated with neighboring Brillouin zones. For example, at momentum $\mathbf{Q}_{1,a}^{CDW} = (-0.57, 2.5, 2.5)$, the intensity arises from CDW modulations with wave-vector component $(-0.57, 0.5, 0.5)$ of the $\mathbf{G}_1 = (0, 2, 2)$ zone and with $(-0.57, -0.5, -0.5)$ of the $\mathbf{G}_2 = (0, 3, 3)$ zone, two different representatives of the $\mathbf{q}_1$-modulation. Calculation of the crystal structure factor of UTe$_2$ (see Supplementary Material Text S6) permits comparison with the intensity of the structural Bragg peak at $\mathbf{G}_0 = (0, 1, 1)$ and yields

$$\left(\frac{\mathbf{u} \cdot \mathbf{Q}_{1,a}^{CDW}}{2}\right)^2 = \frac{A(\mathbf{Q}_{1,a}^{CDW}) \cdot V(\mathbf{Q}_{1,a}^{CDW})}{A(\mathbf{G}_0) \cdot V(\mathbf{G}_0)} \cdot \frac{s(\mathbf{G}_0)}{s(\mathbf{G}_1) + s(\mathbf{G}_2)}. \qquad (3)$$

Following this procedure, we first look at the measurements taken at 3.37 keV shown in Section "Survey for putative CDW order in bulk". For resolution-limited CDW peaks maximum atomic displacements can be estimated using the above equation for $V(\mathbf{Q}_{1,a}^{CDW}) \approx V(\mathbf{G}_0)$. Along the (100) direction we determine $u_{\|(100)} \le 4.7 \cdot 10^{-3}\,Å$ and along the (011) direction $u_{\|(011)} \le 1.2 \cdot 10^{-3}\,Å$. In comparison, the atomic displacements in the CDW in YBa$_2$Cu$_3$O$_{6.67}$ are of order $u \approx 3 \cdot 10^{-3}\,Å$[38], or similarly $u \approx 10^{-3}\,Å$ in related compounds[40].

Now we turn to the data shown in Fig. 4, where we searched for long-range CDW order with nanometer-sized correlation lengths. Again using Eq. (3) with consideration of multiple domains, we can find upper bounds for displacement amplitudes of CDWs having widths $f_{CDW} = (f_{min} + f_{max})/2 = 0.09\,Å^{-1}$. Since the CDW peak width is much larger than the REXS momentum resolution, the integration volume is essentially resolution independent: $V(\mathbf{Q}_{CDW}) = (f_{CDW}^2 + f_{res}^2)^{3/2} \approx f_{CDW}^3$. Considering data from Fig. 4b, we find that atomic displacements at 4.95 keV incident energy are limited by $u \le 0.59\,Å$ along the (100) direction and $u \le 0.1\,Å$ along the (011) direction.

While our data set relatively strong lower bounds on absolute intensities of charge order with short correlation lengths, the bounds on maximum displacement amplitudes are relatively weak. The reason here is that short correlation lengths naturally extend over larger regions in momentum space and result, in turn, in relatively small diffraction peak amplitudes ($V(\mathbf{Q}_{1,a}^{CDW}) \gg V(\mathbf{G}_0)$ in Eq. (3)). Therefore, we conclude that finite displacements with small correlation lengths are in agreement with our data, even though the structure factor is below the detection limits.

## Discussion

Our measurements using resonant X-rays show that the charge modulations earlier observed in STM above the SC transition at the surface of UTe$_2$ are absent from the bulk within our detection limits posed by resonant X-ray diffraction. Our measurements were performed at $T = 2.2$ K, about four times lower than the highest temperature the CDW in UTe$_2$ is reported to exist[25]. Our results did not reveal diffraction intensity in the vicinity of the CDW wave vectors at either the U M$_4$ or Te L$_1$ absorption edges above the background level.

The coverage of our fine mesh-grid search near the $\mathbf{q}_1$ CDW wave vector rules out the presence of sharp resolution-limited charge-order peaks. Notably, the maximum diffraction intensity of such a CDW peak at the U M edge is at least five orders of magnitude smaller relative to the intensity of the strongest Bragg peaks observed. Atomic displacements of such modulations are almost four orders of magnitude smaller than the respective atomic units along (100) and (011).

We further investigated reciprocal space with respect to CDW peaks having short correlation lengths in the bulk. Comparing REXS intensities, we find that maximum intensities of such charge order peaks are at least eight orders of magnitude weaker than nearby structural Bragg peaks. We found further that finite atomic displacements of such modulations are not in disagreement with our data, if the correlation length is small enough.

We highlight that neither thermodynamic measurements[29-31] nor inelastic neutron scattering identified any signatures suggestive of bulk charge order above the SC transition[41-44]. In combination with our results, the emergence of a surface charge-density wave in the normal state provides the most consistent picture. As for the SC state, our results point to two possible conclusions: either the density-wave orders condense simultaneously at $T_c$ in the bulk, in which case PDW order is likely the mother phase (cf. ref. 23), or these charge modulations are restricted to the surface. Bulk scattering measurements in the SC state are required to address this outstanding question.

We note that two additional experimental investigations of UTe$_2$ at low temperature–a non-resonant X-ray diffraction measurement[45] as well as pulse echo and resonant ultrasound measurements[46]–have been reported concurrently with this work. Notably, both studies, performed independently and on samples grown by different groups, are in agreement with our results. Our complementary study provides strict bounds on the structure factor posed by the detection limits in our resonant X-ray diffraction experiments.

## Methods

### Sample preparation

For our experiments, a single crystal of UTe$_2$ with a SC transition at $T_C = 1.8$ K (cf. ref. 31), was used. The sample was oriented with the (100) axis horizontal within 2° using Laue backscattering. See Supplemental Material Fig. S3. The sample surface was the (011) plane, with a surface normal 24° away from the (001) direction and 66 deg away from the (010) direction. The sample was mounted to a copper holder using EPOTEK E4110 silver epoxy. The epoxy was allowed to cure for >3 days in an Ar environment. Aluminum cleave pins were glued to the tops of the samples with TorrSeal and allowed to cure for 1 day in an a Ar environment. Prior to measurement the samples were cleaved, then transferred to vacuum to avoid surface contamination and degradation. No oxidation was observed during measurement or upon removal of the sample from the cryostat.

### Resonant elastic X-ray scattering

Experiments were carried out in the second experimental hutch EH2 of beamline P09 at the synchrotron source PETRA III[47]. The X-ray diffraction was carried out in a horizontal scattering geometry. The UTe$_2$ sample was cooled using a variable temperature insert in a cryomagnet able to provide magnetic fields up to 14 T. X-rays with incident linear polarization were chosen using phase plates[48].

We use the conventional orthorhombic basis with lattice parameters $a = 4.16$ Å, $b = 6.13$ Å, and $c = 13.97$ Å for the description of UTe$_2$ crystals. Momentum transfers are denoted by capital bold letters, **Q**, and given in the reference-frame of the three-dimensional Brillouin zone of UTe$_2$ by means of $\mathbf{Q} = H\mathbf{b}_1 + K\mathbf{b}_2 + L\mathbf{b}_3$, where $\mathbf{b}_i = \epsilon_{ijk} 2\pi/V \cdot \mathbf{b}_j \times \mathbf{b}_k$. $H$, $K$, $L$ are the Miller indices and $\epsilon_{ijk}$ is the Levi-Civita symbol and $V$ the volume of the conventional unit cell in real-space. Propagation vectors of bulk long-range order are denoted by lowercase bold letters, **q**. In order to study specific wave-vectors, we considered appropriate representatives from respective wave-vector stars that were accessible with REXS. Momentum transfers of structural Bragg peaks are labeled with the letter **G**.

Characteristic high-symmetry points in the three-dimensional Brillouin zone of UTe$_2$ are denoted by capital letters. Points and coordinates in the two-dimensional reciprocal lattice, used to describe the Fourier transformation of STM images, are given in lowercase letters.

Structural Bragg peaks measured on UTe$_2$ essentially featured Gaussian profiles. Their intensity may be modeled by:

$$G(\mathbf{Q}) := A \cdot E(\mathbf{Q}) \qquad (4)$$

where $A$ denotes the amplitude (or the maximum) and $E$ is a Gaussian profile with full width at half maxima $f_{\mathbf{d}_1}, f_{\mathbf{d}_2}$, and $f_{\mathbf{d}_3}$ along the three directions $\mathbf{d}_1 = (1, 0, 0)$, $\mathbf{d}_2 = (0, 1, 1)$, and $\mathbf{d}_3 = \mathbf{d}_1 \times \mathbf{d}_2$, respectively. Integrated intensity of a Bragg peak is therefore given by $I = \frac{1}{8}\sqrt{\frac{\pi}{\ln(2)}}^3 \cdot A f_{\mathbf{d}_1} f_{\mathbf{d}_2} f_{\mathbf{d}_2}$. The profile of structural Bragg peaks is essentially restricted by resolution. We, therefore, define $f_{\mathbf{d}_1} f_{\mathbf{d}_2} f_{\mathbf{d}_2}$ as the experimental resolution volume in momentum space.

Structure factors of UTe$_2$ for X-ray scattering were calculated with scattering amplitudes provided by the International Tables for Crystallography, ref. 32. Details on these calculations are provided in the Supplementary Information, Text S6.

## Data availability

The datasets generated during and/or analyzed during the current study are available from the corresponding authors on request. The raw diffraction data shown in the manuscript are available at the Zenodo database under accession code digital object identifier: https://zenodo.org/records/13948392[49].

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

## Acknowledgements

W.S. acknowledges fruitful discussions with Aline Ramirez. The authors further thank Simon Gerber for discussions. W.S. was supported through funding from the European Union's Horizon 2020 research and innovation program under the Marie Sklodowska-Curie grant agreement No 884104 (PSI-FELLOW-III-3i). Work at Los Alamos National Laboratory was performed under the auspices of the U.S. Department of Energy, Office of Basic Energy Sciences, Division of Materials Science and Engineering. C.S.K. acknowledges support from the Laboratory Directed Research and Development program. P.A. acknowledges support from the Gordon and Betty Moore Foundation, EPiQS grant GBMF9452. J.C. acknowledges support from Swiss National Science foundation under grant 200021-188564. M.J. acknowledges funding by the Swiss National Science Foundation through the project "Berry-Phase Tuning in Heavy f-Electron Metals" (200650). We acknowledge DESY (Hamburg, Germany), a member of the Helmholtz Association HGF, for the provision of experimental facilities. Parts of this research were carried out at PETRA III at DESY. Beamtime was allocated for proposal I-20221340 EC.

## Author contributions

M.J., P.F.S.R., and W.S. conceived the study. J.C., M.J., and W.S. designed the experiments. P.F.S.R. grew the samples. C.S.K., J.V., S.F., and W.S. carried out the X-ray experiments. C.S.K., P.F.S.R., and W.S. interpreted the results in discussions with M.J., J.C., and P.A. C.K., P.F.S.R., and W.S. wrote the paper with input from all the authors.

## Competing interests

The authors declare no competing interests.
