## [Transparent Peer Review file · Nature Communications]

Absence of bulk charge density wave order in the normal state of UTe_2

Corresponding Author: Dr Wolfgang Simeth

Version 0:

Reviewer comments:

Reviewer #1

(Remarks to the Author)

This work, together with several recently studies posted on arXiv (ref. 44 and 45 of the manuscript), aims to scrutinize the possible CDW and PDW observed on the surface of UTe_2 . To enhance the sensitivity, the measurements were performed at the Te L1 edge and U M4 edge. The absence of CDW superlattice peak at 2.2 K in extended q-space led the authors to conclude: either the density-wave orders condense simultaneously at T_c in the bulk or the charge modulations are restricted to the surface.

This work, and ref. 44 and 45, are in the spirit of comment of the previous STM study. While I don't find major scientific advancement, I strongly feel that these efforts deserve wider visibility in the community as they provide complete picture of controversial physical properties. With this in mind, I in principle, support its publication in Nature Communications.

Just one suggestion: although all bulk measurements consistently support the absence of CDW in UTe_2 above TSC, it may worthwhile to comment on the possibility that the CDW wavevector is different in the bulk. For instance, STM studies of BSCCO show $Q_{\text{CDW}} \sim 0.25$ on the surface. Bulk studies, however, show $Q_{\text{CDW}} \sim 0.3$.

Reviewer #2

(Remarks to the Author)

The manuscript by Kengle et al. reports on the absence of charge density wave (CDW) order in the bulk of UTe_2 , a material known for its spin-triplet superconductivity. Using resonant elastic x-ray scattering (REXS) at both the U M4 and Te L1 edges, they investigated three candidate CDW wave vectors in three-dimensional momentum space, which project onto the surface CDW wave vectors identified in previous STM studies. However, they did not observe any signatures of CDW formation within the detection limits. Based on their results, the authors concluded that, in the normal state, the formation of surface CDW provides the most conclusive picture.

The authors conducted careful REXS measurements, and their conclusions are consistent with recent publications. However, I found it difficult to understand how they derived the "estimated CDW wave vectors" in 3D momentum space from the STM data (Fig. 1B). As STM is a surface-sensitive technique that in this case visualizes electronic density on the (011) surface of UTe_2 , the FFT image is obtained in a 2D momentum space (Fig. 1A). Despite this, the authors "estimated" the corresponding CDW positions in 3D momentum space (Fig. 1B) and limited their REXS search to the vicinity (in H) of these specific wave vectors.

The authors should elaborate on the procedure and assumptions they used to derive the 3D estimations shown in Fig. 1B and C.

Reviewer #3

(Remarks to the Author)

In this manuscript, the authors present the results of resonant elastic X-ray scattering measurements carried out for examining the existence of the CDW order in the bulk of UTe_2 , a spin-triplet superconductor candidate, above the superconducting transition temperature T_c . This study is motivated by the recent STM measurements for the (011) surface of UTe_2 in which a PDW state below T_c and a CDW state above T_c are observed. The clarification of the CDW order in UTe_2 is quite important for understanding the character of the superconducting state, which is, to this date, the subject of much debate.

In this REXS study, the precise scanning around the wave numbers for which the CDW is detected via the previous STM

measurements is performed, and it is found that the CDW order is absent in the bulk within the detection limit. On the basis of this observation, the authors conclude that the CDW order detected in the STM measurements is restricted to the surface, or the bulk CDW exists only below T_c . The experimental data shown in FIGs. 3 and 4 which indicate the absence of characteristic peak structures are persuasive, supporting the main claim. The results of this study provide crucially important informations for revealing the nature of the CDW and PDW orders reported in the STM studies, accelerating the future investigations toward the full understanding of the pairing state of UTe₂. Because of the above reasons, I would like to strongly recommend the manuscript for publication in Nature Communications.

A minor point:

there are some typos in the manuscript:

e.g. in the 8th lines on page 8, "L₀ coordinates in (B)" should be read as "L₀ coordinates in (A)".

I would like to recommend the authors to check the manuscript carefully before the final process of publication.

Point-To-Point Replies on the reviewers' comments

We thank the three referees for their efforts to review our manuscript. In the following, we present detailed point-to-point replies, where the reviewers' comments are shown in black and our replies are shown in blue. Upon the suggestions of the three referees, we improved our manuscript with tiny modifications. The revised manuscript is attached. For the sake of clarity, we attached a second manuscript version in that all relevant changes are shown in blue color.

Reviewer #1 (Remarks to the Author):

This work, together with several recently studies posted on arXiv (ref. 44 and 45 of the manuscript), aims to scrutinize the possible CDW and PDW observed on the surface of UTe₂. To enhance the sensitivity, the measurements were performed at the Te L1 edge and U M4 edge. The absence of CDW superlattice peak at 2.2 K in extended q-space led the authors to conclude: either the density-wave orders condense simultaneously at T_c in the bulk or the charge modulations are restricted to the surface.

We thank the first reviewer for this summary of our work.

This work, and ref. 44 and 45, are in the spirit of comment of the previous STM study. While I don't find major scientific advancement, I strongly feel that these efforts deserve wider visibility in the community as they provide complete picture of controversial physical properties. With this in mind, I in principle, support its publication in Nature Communications.

We thank the reviewer for their assessment and are pleased to hear that they support publication in Nature Communications, emphasizing the importance to make the results visible in the community.

Just one suggestion: although all bulk measurements consistently support the absence of CDW in UTe₂ above TSC, it may worthwhile to comment on the possibility that the CDW wavevector is different in the bulk. For instance, STM studies of BSCCO show $Q_{CDW} \sim 0.25$ on the surface. Bulk studies, however, show $Q_{CDW} \sim 0.3$.

We thank the first referee for this suggestion. We included a remark in our manuscript about the extended H -ranges that we used in order to account for differences of wave-vectors in the bulk.

Reviewer #2 (Remarks to the Author):

The manuscript by Kengle et al. reports on the absence of charge density wave (CDW) order in the bulk of UTe_2 , a material known for its spin-triplet superconductivity. Using resonant elastic x-ray scattering (REXS) at both the U M4 and Te L1 edges, they investigated three candidate CDW wave vectors in three-dimensional momentum space, which project onto the surface CDW wave vectors identified in previous STM studies. However, they did not observe any signatures of CDW formation within the detection limits. Based on their results, the authors concluded that, in the normal state, the formation of surface CDW provides the most conclusive picture.

We thank the second reviewer for this summary.

The authors conducted careful REXS measurements, and their conclusions are consistent with recent publications. However, I found it difficult to understand how they derived the "estimated CDW wave vectors" in 3D momentum space from the STM data (Fig. 1B). As STM is a surface-sensitive technique that in this case visualizes electronic density on the (011) surface of UTe_2 , the FFT image is obtained in a 2D momentum space (Fig. 1A). Despite this, the authors "estimated" the corresponding CDW positions in 3D momentum space (Fig. 1B) and limited their REXS search to the vicinity (in H) of these specific wave vectors.

The authors should elaborate on the procedure and assumptions they used to derive the 3D estimations shown in Fig. 1B and C.

We thank the second referee for their assessment as well as for this suggestion. Our search for charge-density wave peaks followed a careful procedure based on a well-considered calculation of wave-vectors. We are sorry that our manuscript did not convey that correctly. In order to improve this part of the presentation, we made modifications in the manuscript and inserted a paragraph in the Supplementary information.

Reviewer #3 (Remarks to the Author):

In this manuscript, the authors present the results of resonant elastic X-ray scattering measurements carried out for examining the existence of the CDW order in the bulk of UTe₂, a spin-triplet superconductor candidate, above the superconducting transition temperature T_c . This study is motivated by the recent STM measurements for the (011) surface of UTe₂ in which a PDW state below T_c and a CDW state above T_c are observed. The clarification of the CDW order in UTe₂ is quite important for understanding the character of the superconducting state, which is, to this date, the subject of much debate.

In this REXS study, the precise scanning around the wave numbers for which the CDW is detected via the previous STM measurements is performed, and it is found that the CDW order is absent in the bulk within the detection limit.

On the basis of this observation, the authors conclude that the CDW order detected in the STM measurements is restricted to the surface, or the bulk CDW exists only below T_c .

We thank the reviewer for their summary of our work.

The experimental data shown in FIGs. 3 and 4 which indicate the absence of characteristic peak structures are persuasive, supporting the main claim.

The results of this study provide crucially important informations for revealing the nature of the CDW and PDW orders reported in the STM studies, accelerating the future investigations toward the full understanding of the pairing state of UTe₂.

Because of the above reasons, I would like to strongly recommend the manuscript for publication in Nature Communications.

We thank the third referee for their assessment and are delighted to hear that they support publication in Nature Communications.

A minor point:

there are some typos in the manuscript:

e.g. in the 8th lines on page 8, “L_0 coordinates in (B)” should be read as “L_0 coordinates in (A)”.

I would like to recommend the authors to check the manuscript carefully before the final process of publication.

We apologize for typos in the manuscript and are thankful for this recommendation. We went through the text again carefully to avoid further typos.